# Universal consistency and minimax rates for online Mondrian Forests

**Jaouad Mourtada**
Centre de Mathématiques Appliquées
École Polytechnique, Palaiseau, France
jaouad.mourtada@polytechnique.edu

**Stéphane Gaïffas**
Centre de Mathématiques Appliquées
École Polytechnique,Palaiseau, France
stéphane.gaiffas@polytechnique.edu

**Erwan Scornet**
Centre de Mathématiques Appliquées
École Polytechnique,Palaiseau, France
erwan.scornet@polytechnique.edu

## Abstract

We establish the consistency of an algorithm of Mondrian Forests [LRT14, LRT16], a randomized classification algorithm that can be implemented online. First, we amend the original Mondrian Forest algorithm proposed in [LRT14], that considers a *fixed* lifetime parameter. Indeed, the fact that this parameter is fixed hinders the statistical consistency of the original procedure. Our modified Mondrian Forest algorithm grows trees with increasing lifetime parameters $\lambda_n$, and uses an alternative updating rule, allowing to work also in an online fashion. Second, we provide a theoretical analysis establishing simple conditions for consistency. Our theoretical analysis also exhibits a surprising fact: our algorithm achieves the minimax rate (optimal rate) for the estimation of a Lipschitz regression function, which is a strong extension of previous results [AG14] to an *arbitrary dimension*.

## 1 Introduction

Random Forests (RF) are state-of-the-art classification and regression algorithms that proceed by averaging the forecasts of a number of randomized decision trees grown in parallel (see [Bre01, Bre04, GEW06, BDL08, Bia12, BS16, DMdF14, SBV15]). Despite their widespread use and remarkable success in practical applications, the theoretical properties of such algorithms are still not fully understood [Bia12, DMdF14]. Among these methods, *purely random forests* [Bre00, BDL08, Gen12, AG14] that grow the individual trees independently of the sample, are particularly amenable to theoretical analysis; the consistency of such classifiers was obtained in [BDL08].

An important limitation of the most commonly used random forests algorithms, such as Breiman's Random Forest [Bre01] and the Extra-Trees algorithm [GEW06], is that they are typically trained in a batch manner, using the whole dataset to build the trees. In order to enable their use in situations when large amounts of data have to be incorporated in a streaming fashion, several online adaptations of the decision trees and RF algorithms have been proposed [DH00, TGP11, SLS$^+$09, DMdF13].

Of particular interest in this article is the *Mondrian Forest* algorithm, an efficient and accurate online random forest classifier [LRT14]. This algorithm is based on the Mondrian process [RT09, Roy11], a natural probability distribution on the set of recursive partitions of the unit cube $[0, 1]^d$. An appealing property of Mondrian processes is that they can be updated in an online fashion: in [LRT14], the use of the *conditional Mondrian* process enabled to design an online algorithm that matched its batch counterpart. While Mondrian Forest offer several advantages, both computational and in terms of

predictive performance, the algorithm proposed in [LRT14] depends on a *fixed* lifetime parameter $\lambda$ that guides the complexity of the trees. Since this parameter has to be set in advance, the resulting algorithm is inconsistent, as the complexity of the randomized trees remains bounded. Furthermore, an analysis of the learning properties of Mondrian Forest – and in particular of the influence and proper theoretical tuning of the lifetime parameter $\lambda$ – is still lacking.

In this paper, we propose a modified online random forest algorithm based on Mondrian processes. Our algorithm retains the crucial property of the original method [LRT14] that the decision trees can be updated incrementally. However, contrary to the original approach, our algorithm uses an increasing sequence of lifetime parameters $(\lambda_n)_{n \geqslant 1}$, so that the corresponding trees are increasingly complex, and involves an alternative online updating algorithm. We study such classification rules theoretically, establishing simple conditions on the sequence $(\lambda_n)_{n \geqslant 1}$ to achieve consistency, see Theorem 1 from Section 5 below.

In fact, Mondrian Forests achieve much more than what they were designed for: while they were primarily introduced to derive an online algorithm, we show in Theorem 2 (Section 6) that they actually achieve minimax convergence rates for Lipschitz conditional probability (or regression) functions in arbitrary dimension. To the best of our knowledge, such results have only been proved for very specific purely random forests, where the covariate dimension is equal to one.

**Related work.** While random forests were introduced in the early 2000s [Bre01], as noted by [DMdF14] the theoretical analysis of these methods is outpaced by their practical use. The consistency of various simplified random forests algorithms is first established in [BDL08], as a byproduct of the consistency of individual tree classifiers. A recent line of research [Bia12, DMdF14, SBV15] has sought to obtain theoretical guarantees (*i.e.* consistency) for random forests variants that more closely resembled the algorithms used in practice. Another aspect of the theoretical study of random forests is the bias-variance analysis of simplified versions of random forests [Gen12, AG14], such as the *purely random forests* (PRF) model that performs splits independently of the data. In particular, [Gen12] shows that some PRF variants achieve the minimax rate for the estimation of a Lipschitz regression functions in dimension 1. Additionally, the bias-variance analysis is extended in [AG14], showing that PRF can also achieve minimax rates for $C^2$ regression functions in dimension one, and considering higher dimensional models of PRF that achieve suboptimal rates.

Starting with [SLS+09], online variants of the random forests algorithm have been considered. In [DMdF13], the authors propose an online random forest algorithm and prove its consistency. The procedure relies on a partitioning of the data into two streams: a *structure stream* (used to grow the tree structure) and an *estimation stream* (used to compute the prediction in each leaf). This separation of the data into separate streams is a way of simplifying the proof of consistency, but leads to a non-realistic setting in practice.

A major development in the design of online random forests is the introduction of the *Mondrian Forest* (MF) classifier [LRT14, LRT16]. This algorithm makes an elegant use of the *Mondrian Process*, introduced in [RT09], see also [Roy11, OR15], to draw random trees. Indeed, this process provides a very convenient probability distribution over the set of recursive, tree-based partitions of the hypercube. In [BLG+16], the links between the Mondrian process and the Laplace kernel are used to design random features in order to efficiently approximate kernel ridge regression, leading to the so-called *Mondrian kernel* algorithm.

Our approach differs from the original Mondrian Forest algorithm [LRT14], since it introduces a "dual" construction, that works in the "time" domain (lifetime parameters) instead of the "space" domain (features range). Indeed, in [LRT14], the splits are selected using a Mondrian process on the range of previously observed features vectors, and the online updating of the trees is enabled by the possibility of extending a Mondrian process to a larger cell using *conditional Mondrian* processes. Our algorithm incrementally grows the trees by extending the lifetime; the online update of the trees exploits the Markov property of the Mondrian process, a consequence of its formulation in terms of competing exponential clocks.

## 2   Setting and notation

We first explain the considered setting allowing to state consistency of our procedure, and we describe and set notation for the main concepts used in the paper, namely trees, forests and partitions.

**Considered setting.** Assume we are given an i.i.d. sequence $(X_1, Y_1), (X_2, Y_2) \ldots$ of $[0,1]^d \times \{0,1\}$-valued random variables that come sequentially, such that each $(X_i, Y_i)$ has the same distribution as $(X, Y)$. This unknown distribution is characterized by the distribution $\mu$ of $X$ on $[0,1]^d$ and the conditional probability $\eta(x) = \mathbb{P}(Y = 1 \mid X = x)$.

At each time step $n \geqslant 1$, we want to output a 0-1-valued *randomized classification rule* $g_n(\cdot, Z, \mathscr{D}_n) : [0,1]^d \to \{0,1\}$, where $\mathscr{D}_n = ((X_1, Y_1), \ldots, (X_n, Y_n))$ and $Z$ is a random variable that accounts for the randomization procedure; to simplify notation, we will generally denote $\widehat{g}_n(x, Z) = g_n(x, Z, \mathscr{D}_n)$. The quality of a randomized classifier $g_n$ is measured by its probability of error

$$L(g_n) = \mathbb{P}(g_n(X, Z, \mathscr{D}_n) \neq Y \mid \mathscr{D}_n) = \mathbb{P}_{(X,Y),Z}(g_n(X, Z, \mathscr{D}_n) \neq Y) \tag{1}$$

where $\mathbb{P}_{(X,Y),Z}$ denotes the integration with respect to $(X, Y), Z$ alone. The quantity of Equation (1) is minimized by the *Bayes classifier* $g^*(x) = \mathbf{1}_{\{\eta(x) > \frac{1}{2}\}}$, and its loss, the *Bayes error*, is denoted $L^* = L(g^*)$. We say that a sequence of classification rules $(g_n)_{n \geqslant 1}$ is *consistent* whenever $L(g_n) \to L^*$ in probability as $n \to \infty$.

*Remark* 1. We restrict ourselves to binary classification, note however that our results and proofs can be extended to multi-class classification.

**Trees and Forests.** The classification rules $(g_n)_{n \geqslant 1}$ we consider take the form of a *random forest*, defined by averaging randomized tree classifiers. More precisely, let $K \geqslant 1$ be a fixed number of randomized classifiers $\widehat{g}_n(x, Z_1), \ldots, \widehat{g}_n(x, Z_K)$ associated to the same randomized mechanism, where the $Z_k$ are i.i.d. Set $Z^{(K)} = (Z_1, \ldots, Z_K)$. The *averaging classifier* $\widehat{g}_n^{(K)}(x, Z^{(K)})$ is defined by taking the majority vote among the values $g_n(x, Z_k)$, $k = 1, \ldots, K$.

Our individual randomized classifiers are *decision trees*. A decision tree $(T, \Sigma)$ is composed of the following components:

- A finite rooted ordered binary tree $T$, with nodes $\mathcal{N}(T)$, interior nodes $\mathcal{N}^\circ(T)$ and leaves $\mathcal{L}(T)$ (so that $\mathcal{N}(T)$ is the disjoint union of $\mathcal{N}^\circ(T)$ and $\mathcal{L}(T)$). Each interior node $\eta$ has a left child $\texttt{left}(\eta)$ and a right child $\texttt{right}(\eta)$;
- A family of *splits* $\Sigma = (\sigma_\eta)_{\eta \in \mathcal{N}^\circ(T)}$ at each interior node, where each split $\sigma_\eta = (d_\eta, \nu_\eta)$ is characterized by its split dimension $d_\eta \in \{1, \ldots, d\}$ and its threshold $\nu_\eta$.

Each randomized classifier $\widehat{g}_n(x, Z_k)$ relies on a decision tree $T$, the random variable $Z_k$ is the random sampling of the splits $(\sigma_\eta)$ defining $T$. This sampling mechanism, based on the Mondrian process, is defined in Section 3.

We associate to $M = (T, \Sigma)$ a partition $(A_\phi)_{\phi \in \mathcal{L}(T)}$ of the unit cube $[0,1]^d$, called a *tree partition* (or *guillotine partition*). For each node $\eta \in \mathcal{N}(T)$, we define a hyper-rectangular region $A_\eta$ recursively:

- The cell associated to the root of $T$ is $[0,1]^d$;
- For each $\eta \in \mathcal{N}^\circ(T)$, we define

$$A_{\texttt{left}(\eta)} := \{x \in A_\eta : x_{d_\eta} \leqslant \nu_\eta\} \quad \text{and} \quad A_{\texttt{right}(\eta)} := A_\eta \setminus A_{\texttt{left}(\eta)}.$$

The leaf cells $(A_\phi)_{\phi \in \mathcal{L}(T)}$ form a partition of $[0,1]^d$ by construction. In the sequel, we will identify a tree with splits $(T, \Sigma)$ with the associated tree partition $M(T, \Sigma)$, and a node $\eta \in \mathcal{N}(T)$ with the cell $A_\eta \subset [0,1]^d$. The decision tree classifier outputs a constant prediction of the label in each leaf cell $A_\eta$ using a simple majority vote of the labels $Y_i$ $(1 \leqslant i \leqslant n)$ such that $X_i \in A_\eta$.

## 3 A new online Mondrian Forest algorithm

We describe the Mondrian Process in Section 3.1, and recall the original Mondrian Forest procedure in Section 3.2. Our procedure is introduced in Section 3.3.

### 3.1 The Mondrian process

The probability distribution we consider on tree-based partitions of the unit cube $[0,1]^d$ is the Mondrian process, introduced in [RT09]. Given a rectangular box $C = \prod_{j=1}^d [a_j, b_j]$, we denote

$|C| := \sum_{j=1}^{d}(b_j - a_j)$ its *linear dimension*. The Mondrian process distribution $\mathrm{MP}(\lambda, C)$ is the distribution of the random tree partition of $C$ obtained by the sampling procedure $\mathtt{SampleMondrian}(\lambda, C)$ from Algorithm 1.

---

**Algorithm 1** $\mathtt{SampleMondrian}(\lambda, C)$ ; Samples a tree partition distributed as $\mathrm{MP}(\lambda, C)$.

---

1: **Parameters:** A rectangular box $C \subset \mathbf{R}^d$ and a lifetime parameter $\lambda > 0$.
2: **Call** $\mathtt{SplitCell}(C, \tau_C := 0, \lambda)$.

---

---

**Algorithm 2** $\mathtt{SplitCell}(A, \tau, \lambda)$ ; Recursively split a cell $A$, starting from time $\tau$, until $\lambda$

---

1: **Parameters:** A cell $A = \prod_{1 \leqslant j \leqslant d}[a_j, b_j]$, a starting time $\tau$ and a lifetime parameter $\lambda$.
2: **Sample** an exponential random variable $E_A$ with intensity $|A|$.
3: **if** $\tau + E_A \leqslant \lambda$ **then**
4:     **Draw** at random a split dimension $J \in \{1, \ldots, d\}$, with $\mathbb{P}(J = j) = (b_j - a_j)/|A|$, and a split threshold $\nu_J$ uniformly in $[a_J, b_J]$.
5:     **Split** $A$ along the split $(J, \nu_J)$.
6:     **Call** $\mathtt{SplitCell}(\mathtt{left}(A), \tau + E_A, \lambda)$ and $\mathtt{SplitCell}(\mathtt{right}(A), \tau + E_A, \lambda)$.
7: **else**
8:     Do nothing.
9: **end if**

---

## 3.2 Online tree growing: the original scheme

In order to implement an online algorithm, it is crucial to be able to "update" the tree partitions grown at a given time step. The approach of the original Mondrian Forest algorithm [LRT14] uses a slightly different randomization mechanism, namely a Mondrian process supported in the range defined by the past feature points. More precisely, this modification amounts to replacing each call to $\mathtt{SplitCell}(A, \tau, \lambda)$ by a call to $\mathtt{SplitCell}(A^{\mathtt{range}(n)}, \tau, \lambda)$, where $A^{\mathtt{range}(n)}$ is the range of the feature points $X_1, \ldots, X_n$ that fall in $A$ (*i.e.* the smallest box that contains them).

When a new training point $(X_{n+1}, Y_{n+1})$ arrives, the ranges of the training points may change. The online update of the tree partition then relies on the extension properties of the Mondrian process: given a Mondrian partition $M_1 \sim \mathrm{MP}(\lambda, C_1)$ on a box $C_1$, it is possible to efficiently sample a Mondrian partition $M_0 \sim \mathrm{MP}(\lambda, C_0)$ on a larger box $C_0 \supset C_1$ that restricts to $M_1$ on the cell $C_1$ (this is called a "conditional Mondrian", see [RT09]).

*Remark* 2. In [LRT14] a lifetime parameter $\lambda = \infty$ is actually used in experiments, which essentially amounts to growing the trees completely, until the leaves are homogeneous. We will not analyze this variant here, but this illustrates the problem of using a fixed, finite budget $\lambda$ in advance.

## 3.3 Online tree growing: a dual approach

An important limitation of the original scheme is the fact that it requires to fix the lifetime parameter $\lambda$ in advance. In order to obtain a consistent algorithm, it is required to grow increasingly complex trees. To achieve this, we propose to adopt a "dual" point of view: instead of using a Mondrian process with fixed lifetime on a domain that changes as new data points are added, we use a Mondrian process on a fixed domain (the cube $[0, 1]^d$) but with a varying lifetime $\lambda_n$ that grows with the sample size $n$. The rationale is that, as more data becomes available, the classifiers should be more complex and precise. Since the lifetime, rather than the domain, is the parameter that guides the complexity of the trees, it should be this parameter that dynamically adapts to the amount of training data.

It turns out that in this approach, quite surprisingly, the trees can be updated incrementally, leading to an online algorithm. The ability to extend a tree partition $M_{\lambda_n} \sim \mathrm{MP}(\lambda_n, [0, 1]^d)$ into a finer tree partition $M_{\lambda_{n+1}} \sim \mathrm{MP}(\lambda_{n+1}, [0, 1]^d)$ relies on a different property of the Mondrian process, namely the fact that for $\lambda < \lambda'$, it is possible to efficiently sample a Mondrian tree partition $M_{\lambda'} \sim \mathrm{MP}(\lambda', C)$ given its *pruning* $M_\lambda \sim \mathrm{MP}(\lambda, C)$ at time $\lambda$ (obtained by dropping all splits of $M_{\lambda'}$ performed at a time $\tau > \lambda$).

The procedure $\mathtt{ExtendMondrian}(M_\lambda, \lambda, \lambda')$ from Algorithm 3 extends a Mondrian tree partition $M_\lambda \sim \mathrm{MP}(\lambda, C)$ to a tree partition $M_{\lambda'} \sim \mathrm{MP}(\lambda', C)$. Indeed, for each leaf cell $A$ of $M_\lambda$, the fact

---

**Algorithm 3** ExtendMondrian($M_\lambda, \lambda, \lambda'$) ; Extend $M_\lambda \sim \mathrm{MP}(\lambda, C)$ to $M_{\lambda'} \sim \mathrm{MP}(\lambda', C)$

---
1: **Parameters:** A tree partition $M_\lambda$, and lifetimes $\lambda \leqslant \lambda'$.
2: **for** $A$ in $\mathcal{L}(M_\lambda)$ **do**
3:     **Call** SplitCell($A, \lambda, \lambda'$)
4: **end for**

---

that $A$ is a leaf of $M_\lambda$ means that during the sampling of $M_\lambda$, the time of the next candidate split $\tau + E_A$ (where $\tau$ is the time $A$ was formed and $E_A \sim \mathrm{Exp}(|A|)$) was strictly larger than $\lambda$. Now in the procedure ExtendMondrian($M_\lambda, \lambda, \lambda'$), the time of the next candidate split is $\lambda + E'_A$, where $E'_A \sim \mathrm{Exp}(|A|)$. This is precisely the where the trick resides: by the memory-less property of the exponential distribution, the distribution of $\tau_A + E_A$ conditionally on $E_A > \lambda - \tau_A$ is the same as that of $\lambda + E'_A$. The procedure ExtendMondrian can be replaced by the following more efficient implementation:

- Time of the next split of the tree is sampled as $\lambda + E_{M_\lambda}$ with $E_{M_\lambda} \sim \mathrm{Exp}(\sum_{\phi \in \mathcal{L}(M_\lambda)} |A_\phi|)$;

- Leaf to split is chosen using a top-down path from the root of the tree, where the choice between left or right child for each interior node is sampled at random, proportionally to the linear dimension of all the leaves in the subtree defined by the child.

*Remark* 3. While we consider Mondrian partitions on the fixed domain $[0, 1]^d$, our increasing lifetime trick can be used *in conjunction* with a varying domain based on the range of the data (as in the original MF algorithm), simply by applying ExtendMondrian($M_{\lambda_n}, \lambda_n, \lambda_{n+1}$) after having extended the Mondrian to the new range. In order to keep the analysis tractable and avoid unnecessary complications in the analysis, we will study the procedure on a fixed domain only.

Given an increasing sequence $(\lambda_n)_{n \geqslant 1}$ of lifetime parameters, our modified MF algorithm incrementally updates the trees $M_\lambda^{(k)}$ for $k = 1, \ldots, K$ by calling ExtendMondrian($M_{\lambda_n}^{(k)}, \lambda_n, \lambda_{n+1}$), and combines the forecasts of the given trees, as explained in Algorithm 4.

---

**Algorithm 4** MondrianForest($K, (\lambda_n)_{n \geqslant 1}$) ; Trains a Mondrian Forest classifier.

---
1: **Parameters:** The number of trees $K$ and the lifetime sequence $(\lambda_n)_{n \geqslant 1}$.
2: **Initialization:** Start with $K$ trivial partitions $M_{\lambda_0}^{(k)}$, $\lambda_0 := 0$, $k = 1, \ldots, K$. Set the counts of the training labels in each cell to 0, and the labels e.g. to 0.
3: **for** $n = 1, 2, \ldots$ **do**
4:     **Receive** the training point $(X_n, Y_n)$.
5:     **for** $k = 1, \ldots, K$ **do**
6:         **Update** the counts of 0 and 1 (depending on $Y_n$) in the leaf cell of $X_n$ in $M_{\lambda_n}$.
7:         **Call** ExtendMondrian($M_{\lambda_{n-1}}^{(k)}, \lambda_{n-1}, \lambda_n$).
8:         **Fit** the newly created leaves.
9:     **end for**
10: **end for**

---

For the prediction of the label given a new feature vector, our algorithm uses a majority vote over the predictions given by all $K$ trees. However, other choices are possible. For instance, the original Mondrian Forest algorithm [LRT14] places a hierarchical Bayesian prior over the label distribution on each node of the tree, and performs approximate posterior inference using the so-called interpolated Kneser-Ney (IKN) smoothing. Another possibility, that will be developed in an extended version of this work, is tree expert aggregation methods, such as the Context-Tree Weighting (CTW) algorithm [WST95, HS97] or specialist aggregation methods [FSSW97] over the nodes of the tree, adapting them to increasingly complex trees.

Our modification of the original Mondrian Forest replaces the process of online tree growing with a fixed lifetime by a new process, that allows to increase lifetimes. This modification not only allows to prove consistency, but more surprisingly leads to an optimal estimation procedure, in terms of minimax rates, as illustrated in Sections 5 and 6 below.

# 4 Mondrian Forest with fixed lifetime are inconsistent

We state in Proposition 1 the inconsistency of fixed-lifetime Mondrian Forests, such as the original algorithm [LRT14]. This negative result justifies our modified algorithm based on an increasing sequence of lifetimes $(\lambda_n)_{n \geqslant 1}$.

**Proposition 1.** *The Mondrian Forest algorithm (Algorithm 4) with a fixed lifetime sequence $\lambda_n = \lambda$ is inconsistent: there exists a distribution of $(X, Y) \in [0, 1] \times \{0, 1\}$ such that $L^* = 0$ and $L(g_n) = \mathbb{P}(g_n(X) \neq Y)$ does not tend to $0$. This result also holds true for the original Mondrian Forest algorithm with lifetime $\lambda$.*

Proposition 1 is established in Appendix C. The proof uses a result of independent interest (Lemma 3), which states that asymptotically over the sample size, for fixed $\lambda$, the restricted domain does not affect the randomization procedure.

# 5 Consistency of Mondrian Forest with lifetime sequence $(\lambda_n)$

The consistency of the Mondrian Forest used with a properly tuned sequence $(\lambda_n)$ is established in Theorem 1 below.

**Theorem 1.** *Assume that $\lambda_n \to \infty$ and that $\lambda_n^d / n \to 0$. Then, the online Mondrian Forest described in Algorithm 4 is consistent.*

This consistency result is universal, in the sense that it makes no assumption on the distribution of $X$ nor on the conditional probability $\eta$. This contrasts with some consistency results on Random forests, such as Theorem 1 of [DMdF13], which assumes that the density of $X$ is bounded by above and below.

Theorem 1 does not require an assumption on $K$ (number of trees). It is well-known for batch Random Forests that this meta-parameter is not a sensitive tuning parameter, and that it suffices to choose it large enough to obtain good accuracy. The only important parameter is the sequence $(\lambda_n)$, that encodes the complexity of the trees. Requiring an assumption on this meta-parameter is natural, and confirmed by the well-known fact that the tree-depth is the most important tuning parameter for batch Random Forests, see for instance [BS16].

The proof of Theorem 1 can be found in the supplementary material (see Appendix D). The core of the argument lies in two lemmas describing two novel properties of Mondrian trees. Lemma 1 below provides an upper bound of order $O(\lambda^{-1})$ on the diameter of the cell $A_\lambda(x)$ of a Mondrian partition $M_\lambda \sim \mathrm{MP}(\lambda, [0, 1]^d)$. This is the key to control the bias of Mondrian Forests with lifetime sequence that tend to infinity.

**Lemma 1** (Cell diameter)**.** *Let $x \in [0, 1]^d$, and let $D_\lambda(x)$ be the $\ell^2$-diameter of the cell containing $x$ in a Mondrian partition $M_\lambda \sim \mathrm{MP}(\lambda, [0, 1]^d)$. If $\lambda \to \infty$, then $D_\lambda(x) \to 0$ in probability. More precisely, for every $\delta, \lambda > 0$, we have*

$$\mathbb{P}(D_\lambda(x) \geqslant \delta) \leqslant d \left( 1 + \frac{\lambda \delta}{\sqrt{d}} \right) \exp \left( -\frac{\lambda \delta}{\sqrt{d}} \right) \qquad (2)$$

*and*

$$\mathbb{E}\big[ D_\lambda(x)^2 \big] \leqslant \frac{4d}{\lambda^2}. \qquad (3)$$

The proof of Lemma 1 is provided in the supplementary material (see Appendix A). The second important property needed to carry out the analysis is stated in Lemma 2 and helps to control the "variance" of Mondrian forests. It consists in an upper bound of order $O(\lambda^d)$ on the total number of splits performed by a Mondrian partition $M_\lambda \sim \mathrm{MP}(\lambda, [0, 1]^d)$. This ensures that enough data points fall in each cell of the tree, so that the labels of the tree are well estimated. The proof of Lemma 2 is to be found in the supplementary material (see Appendix B).

**Lemma 2** (Number of splits)**.** *If $K_\lambda$ denotes the number of splits performed by a Mondrian tree partition $M_\lambda \sim \mathrm{MP}(\lambda, [0, 1]^d)$, we have $\mathbb{E}(K_\lambda) \leqslant (e(\lambda + 1))^d$.*

*Remark* 4. It is worth noting that controlling the total number of splits ensures that the cell $A_{\lambda_n}(X)$ in which a new random $X \sim \mu$ ends up contains enough training points among $X_1, \dots, X_n$

(see Lemma 4 in appendix D). This enables to get a distribution-free consistency result. Another approach consists in lower-bounding the volume $V_{\lambda_n}(x)$ of $A_{\lambda_n}(x)$ in probability for any $x \in [0,1]^d$, which shows that the cell $A_{\lambda_n}(x)$ contains enough training points, but this would require the extra assumption that the density of $X$ is lower-bounded.

Remarkably, owing to the nice restriction properties of the Mondrian process, Lemmas 1 and 2 essentially provide matching upper and lower bounds on the complexity of the partition. Indeed, in order to partition the cube $[0,1]^d$ in cells of diameter $O(1/\lambda)$, at least $\Theta(\lambda^d)$ cells are needed; Lemma 2 shows that the Mondrian partition in fact contains only $O(\lambda^d)$ cells.

## 6  Minimax rates over the class of Lipschitz functions

The estimates obtained in Lemmas 1 and 2 are quite explicit and sharp in their dependency on $\lambda$, and allow to study the convergence rate of our algorithm. Indeed, it turns out that our modified Mondrian Forest, when properly tuned, can achieve the minimax rate in classification over the class of Lipschitz functions (see e.g. Chapter I.3 in [Nem00] for details on minimax rates). We provide two results: a convergence rate for the estimation of the conditional probabilities, measured by the quadratic risk, see Theorem 2, and a control on the distance between the classification error of our classifier and the Bayes error, see Theorem 3. We provide also similar minimax bounds for the regression setting instead of the classification one in the supplementary material, see Proposition 4 in Appendix E.

Let $\widehat{\eta}_n$ be the estimate of the conditional probability $\eta$ based on the Mondrian Forest (see Algorithm 4) in which:

$(i)$  Each leaf label is computed as the proportion of $1$ in the corresponding leaf;

$(ii)$  Forest prediction results from the average of tree estimates instead of a majority vote.

**Theorem 2.** *Assume that the conditional probability function $\eta : [0,1]^d \to [0,1]$ is Lipschitz on $[0,1]^d$. Let $\widehat{\eta}_n$ be a Mondrian Forest as defined in Points (i) and (ii), with a lifetimes sequence that satisfies $\lambda_n \asymp n^{1/(d+2)}$. Then, the following upper bound holds*

$$\mathbb{E}(\eta(X) - \widehat{\eta}_n(X))^2 = O(n^{-2/(d+2)}) \tag{4}$$

*for $n$ large enough, which correspond to the minimax rate over the set of Lipschitz functions.*

To the best of our knowledge, Theorem 2 is the first to exhibit the fact that a classification method based on a purely random forest can be minimax optimal in an arbitrary dimension. The same kind of result is stated for regression estimation in the supplementary material (see Proposition 4 in Appendix E).

Minimax rates, but only for $d = 1$, were obtained in [Gen12, AG14] for models of purely random forests such as Toy-PRF (where the individual partitions corresponded to randomly shifts of the regular partition of $[0,1]$ in $k$ intervals) and PURF (Purely Uniformly Random Forests, where the partitions were obtained by drawing $k$ random thresholds at random in $[0,1]$).

However, for $d = 1$, tree partitions reduce to partitions of $[0,1]$ in intervals, and do not possess the recursive structure that appears in higher dimensions and makes their precise analysis difficult. For this reason, the analysis of purely random forests for $d > 1$ has typically produced sub-optimal results: for example, [BDL08] show consistency for UBPRF (Unbalanced Purely Random Forests, that perform a fixed number of splits and randomly choose a leaf to split at each step), but with no rate of convergence. A further step was made by [AG14], who studied the BPRF (Balanced Purely Random Forests algorithm, where all leaves were split, so that the resulting tree was complete), and obtained suboptimal rates. In our approach, the convenient properties of the Mondrian process enable to bypass the inherent difficulties met in previous attempts, thanks to its recursive structure, and allow to obtain the minimax rate with transparent proof.

Now, note that the Mondrian Forest classifier corresponds to the plugin classifier $\widehat{g}_n(x) = \mathbf{1}_{\{\widehat{\eta}_n(x) > 1/2\}}$, where $\widehat{\eta}_n$ is defined in Points (i) and (ii). A general theorem (Theorem 6.5 in [DGL96]) allows us to derive upper bounds on the distance between the classification error of $\widehat{g}_n$ and the Bayes error, thanks to Theorem 2.

**Theorem 3.** *Under the same assumptions as in Theorem 2, the Mondrian Forest classifier $\widehat{g}_n$ with lifetime sequence $\lambda_n \asymp n^{1/(d+2)}$ satisfies*

$$L(\widehat{g}_n) - L^* = o(n^{-1/(d+2)}). \tag{5}$$

The rate of convergence $o(n^{-1/(d+2)})$ for the error probability with a Lipschitz conditional probability $\eta$ turns out to be optimal, as shown by [Yan99]. Note that faster rates can be achieved in classification under low noise assumptions such as the *margin assumption* [MT99] (see e.g. [Tsy04, AT07, Lec07]). Such specializations of our results are to be considered in a future work, the aim of the present paper being an emphasis on the appealing optimal properties of our modified Mondrian Forest.

## 7 Experiments

We now turn to the empirical evaluation of our algorithm, and examine its predictive performance (test error) as a function of the training size. More precisely, we compare the modified Mondrian Forest algorithm (Algorithm 4) to batch (Breiman RF [Bre01], Extra-Trees-1 [GEW06]) and online (the Mondrian Forest algorithm [LRT14] with fixed lifetime parameter $\lambda$) Random Forests algorithms. We compare the prediction accuracy (on the test set) of the aforementioned algorithms trained on varying fractions of the training data from $10\%$ to $100\%$.

Regarding our choice of competitors, we note that Breiman's RF is well-established and known to achieve state-of-the-art performance. We also included the *Extra-Trees*-1 (ERT-1) algorithm [GEW06], which is most comparable to the Mondrian Forest classifier since it also draws splits randomly (we note that the ERT-$k$ algorithm [GEW06] with the default tuning $k = \sqrt{d}$ in the `scikit-learn` implementation [PVG$^+$11] achieves scores very close to those of Breiman's RF).

In the case of online Mondrian Forests, we included our modified Mondrian Forest classifier with an increasing lifetime parameter $\lambda_n = n^{1/(d+2)}$ tuned according to the theoretical analysis (see Theorem 3), as well as a Mondrian Forest classifier with constant lifetime parameter $\lambda = 2$. Note that while a higher choice of $\lambda$ would have resulted in a performance closer to that of the modified version (with increasing $\lambda_n$), our inconsistency result (Proposition 1) shows that its error would eventually stagnate given more training samples. In both cases, the splits are drawn within the range of the training feature, as in the original Mondrian Forest algorithm. Our results are reported in Figure 1.

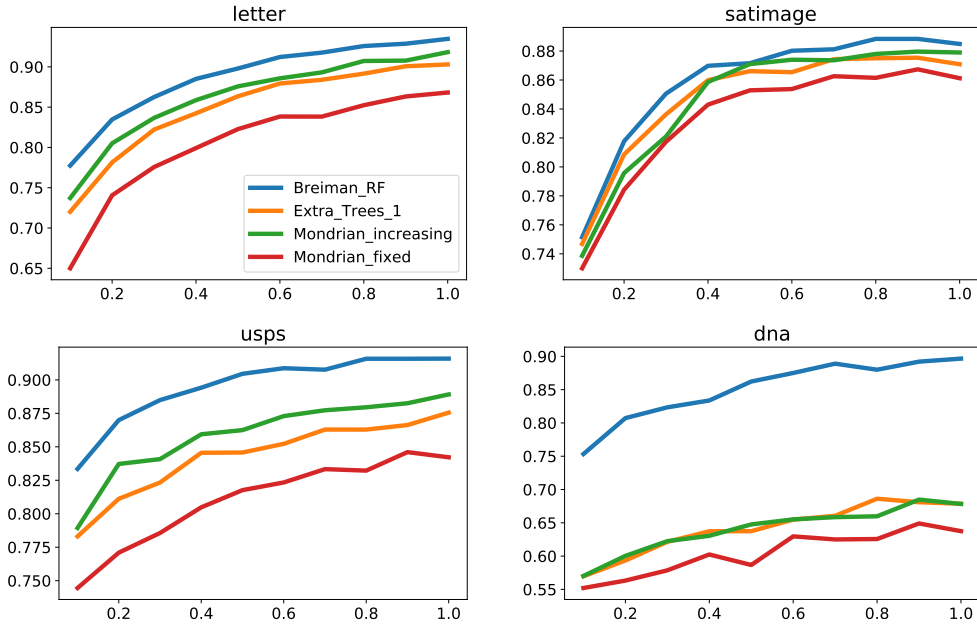

Figure 1: Prediction accuracy as a function of the fraction of data used on several datasets. Modified MF (Algorithm 4) outperforms MF with a constant lifetime, and is better than the batch ERT-1 algorithm. It also performs almost as well as Breiman's RF (a batch algorithm that uses the whole training dataset in order to choose each split) on several datasets, while being incremental and much faster to train. On the *dna* dataset, as noted in [LRT14], Breiman's RF outperforms the other algorithms because of the presence of a large number of irrelevant features.

# 8 Conclusion and future work

Despite their widespread use in practice, the theoretical understanding of Random Forests is still incomplete. In this work, we show that amending the Mondrian Forest classifier, originally introduced to provide an efficient online algorithm, leads to an algorithm that is not only consistent, but in fact minimax optimal for Lipschitz conditional probabilities in arbitrary dimension. This new result suggests promising improvements in the understanding of random forests methods.

A first, natural extension of our results, that will be addressed in a future work, is the study of the rates for smoother regression functions. Indeed, we conjecture that through a more refined study of the local properties of the Mondrian partitions, it is possible to describe exactly the distribution of the cell of a given point. In the spirit of the work of [AG14] in dimension one, this could be used to show improved rates for the bias of forests (e.g. for $C^2$ regression functions) compared to the tree bias, and hence give some theoretical insight to the empirically well-known fact that a forest performs better than individual trees.

Second, the optimal upper bound $O(n^{-1/(d+2)})$ obtained in this paper is very slow when the number of features $d$ is large. This comes from the well-known curse of dimensionality phenomenon, a problem affecting all fully nonparametric algorithms. A standard technique used in high-dimensional settings is to work under a sparsity assumption, where only $s \ll d$ features are informative (*i.e.* affect the distribution of $Y$). In such settings, a natural strategy is to select the splits using the labels $Y_1, \ldots, Y_n$, as most variants of Random Forests used in practice do. For example, it would be interesting to combine a Mondrian process-based randomization with a choice of the best split among several candidates, as performed by the Extra-Tree algorithm [GEW06]. Since the Mondrian Forest guarantees minimax rates, we conjecture that it should improve feature selection of batch random forest methods, and improve the underlying randomization mechanism of these algorithms. From a theoretical perspective, it could be interesting to see how the minimax rates obtained here could be coupled with results on the ability of forests to select informative variables, see for instance [SBV15].

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
