[Supplementary Material]

# Supplementary material for **Universal consistency and minimax rates for online Mondrian Forests**

J. Mourtada, S. Gaïffas and E. Scornet

## A    Proof of Lemma 1: diameter of the cells

We start by recalling some important properties of the Mondrian process, which are exposed in [RT09].

**Fact 1** (Consistency, Mondrian slices). *Let $M_\lambda \sim \mathrm{MP}(\lambda, [0,1]^d)$ be a Mondrian partition, and $C = \prod_{j=1}^d [a_j, b_j] \subset [0,1]^d$, be an axis-aligned box (we authorize lower-dimensional boxes when $a_j = b_j$ for some dimensions $j$). Consider the* restriction $M_\lambda|_C$ *of $M_\lambda$ on $C$, i.e. the partition on $C$ induced by the partition $M_\lambda$ of $[0,1]^d$. Then $M_\lambda|_C \sim \mathrm{MP}(\lambda, C)$.*

**Fact 2** (Dimension 1). *For $d = 1$, the splits from a Mondrian process $M_\lambda \sim \mathrm{MP}(\lambda, [0,1])$ form a subset of $[0,1]$, which is distributed as a Poisson point process of intensity $\lambda dx$.*

We will now establish the technical lemma 1. In what follows, $x \in [0,1]^d$ is arbitrary, and we let $A_\lambda(x)$ denote the (random) cell of a Mondrian partition $M_\lambda \sim \mathrm{MP}(\lambda, [0,1]^d)$ containing $x$.

*Proof of Lemma 1.* Let $A_\lambda(x) = \prod_{j=1}^d [L_\lambda^j(x), R_\lambda^j(x)]$ denote the (random) cell of a Mondrian partition $M_\lambda \sim \mathrm{MP}(\lambda, [0,1]^d)$ containing $x \in [0,1]^d$. By definition, the $\ell^\infty$-diameter $D_\lambda(x)$ of $A_\lambda(x)$ is $\max_{1 \leqslant j \leqslant d}(R_\lambda^j(x) - L_\lambda^j(x))$. Since the random variables $R_\lambda^j(x) - L_\lambda^j(x), 1 \leqslant j \leqslant d$, all have the same distribution (by symmetry of the definition of the Mondrian process with respect to the dimension), it suffices to consider $D_\lambda^1(x) := R_\lambda^1(x) - L_\lambda^1(x)$.

Consider the segment $I^1(x) = [0,1] \times \{(x_j)_{2 \leqslant j \leqslant d}\} \simeq [0,1]$ (through the natural identification) containing $x = (x_j)_{1 \leqslant j \leqslant d}$, and denote $\Phi_\lambda^1(x) \subset [0,1]$ the restriction of $M_\lambda$ to $I^1(x)$. Note that $R_\lambda^1(x)$ (resp. $L_\lambda^1(x)$) is the lowest element of $\Phi_\lambda^1(x)$ that is larger than $x_1$ (resp. the highest element of $\Phi_\lambda^1(x)$ that is smaller than $x_1$), and is equal to 1 (resp. 0) if $\Phi_\lambda^1(x) \cap [x_1, 1]$ (resp. $\Phi_\lambda^1(x) \cap [0, x_1]$) is empty. By the facts 1 and 2, $\Phi_\lambda(x)$ is a Poisson point process of intensity $\lambda$.

Now, note that the characterization of $L_\lambda^1(x)$ and $R_\lambda^1(x)$ in terms of $\Phi_\lambda^1(x)$ (a Poisson process on $[0,1]$) implies the following: the distribution of $(L_\lambda^1(x), R_\lambda^1(x))$ is the same as that of $(\tilde{L}_\lambda^1(x) \vee 0, \tilde{R}_\lambda^1(x) \wedge 1)$, where $\tilde{\Phi}_\lambda^1(x)$ is a Poisson process on $\mathbf{R}$ of intensity $\lambda$, and $\tilde{L}_\lambda^1(x) = \sup(\tilde{\Phi}_\lambda^1(x) \cap (-\infty, x])$, $\tilde{R}_\lambda^1(x) = \inf(\tilde{\Phi}_\lambda^1(x) \cap [x, +\infty))$. By the properties of the Poisson point process, this implies that $(R_\lambda^1(x) - x_1, x_1 - L_\lambda^1(x)) \stackrel{d}{=} (E_1 \wedge (1 - x_1), E_2 \wedge x_1)$, where $E_1, E_2$ are independent exponential random variables with parameter $\lambda$. In particular, $D_\lambda^1(x) = R_\lambda^1(x) - x_1 + x_1 - L_\lambda^1(x)$ is stochastically upper bounded by $E_1 + E_2 \sim \Gamma(2, \lambda)$, so that we have for every $\delta > 0$:

$$\mathbb{P}(D_\lambda^1(x) \geqslant \delta) \leqslant (1 + \lambda\delta)e^{-\lambda\delta} \tag{6}$$

(with equality if $\delta \leqslant x_1 \wedge (1 - x_1)$), and $\mathbb{E}[D_\lambda^1(x)^2] \leqslant \mathbb{E}(E_1^2) + \mathbb{E}(E_2^2) = \frac{4}{\lambda^2}$. Finally, the bound (2) for the diameter $D_\lambda(x) = \sqrt{\sum_{j=1}^d D_\lambda^j(x)^2}$ follows from the observation that $\mathbb{P}(D_\lambda(x) \geqslant \delta) \leqslant \mathbb{P}(\exists j : D_\lambda^j(x) \geqslant \frac{\delta}{\sqrt{d}}) \leqslant d\,\mathbb{P}(D_\lambda^1(x) \geqslant \frac{\delta}{\sqrt{d}})$ and inequality (6); the bound (3) is obtained by noting that $\mathbb{E}[D_\lambda(x)^2] = d\,\mathbb{E}[D_\lambda^1(x)^2] \leqslant \frac{4d}{\lambda^2}$. $\qquad\square$

## B    Proof of Lemma 2: number of splits

*Proof.* Let $A \subset \mathbf{R}^d$ be an arbitrary box, and let $K_\lambda^A$ denote the number of splits performed by $M_\lambda^A \sim \mathrm{MP}(\lambda, A)$. As shown in the proof of Proposition 3 in [BLG+16], since the time until a leaf $\phi$ is split follows an exponential distribution of rate $|A_\phi| \leqslant |A|$ (independently of the other leaves), the number of leaves $K_t + 1 \geqslant K_t$ at time $t$ is dominated by the number of individuals in a Yule process with rate $|A|$, which gives the first estimate

$$\mathbb{E}(K_\lambda^A) \leqslant \exp(\lambda|A|). \tag{7}$$

This bound can be refined to the correct order of magnitude in $\lambda$ in the following way. Consider the covering $\mathscr{C}$ of $A$ by a regular grid of $\lceil\lambda\rceil^d$ boxes obtained by dividing each coordinate of $A$ in $\lceil\lambda\rceil$. Since each split of $A$ induces a split in at least one box $C \in \mathscr{C}$ (*i.e.* a split in the restriction $M_\lambda^C$ of $M_\lambda^A$ to $C$), and since $M_\lambda^C \sim \mathrm{MP}(\lambda, C)$ by Fact 1,

$$\mathbb{E}(K_\lambda^A) \leqslant \sum_{C \in \mathscr{C}} \mathbb{E}(K_\lambda^C) \overset{(*)}{\leqslant} \lceil\lambda\rceil^d \exp\left(\lambda \frac{|A|}{\lceil\lambda\rceil}\right) \leqslant (\lambda+1)^d \exp(|A|) \tag{8}$$

where in the inequality (*) we applied the bound (7) to every cell $C \in \mathscr{C}$ (and the fact that $|C| = |A|/\lceil\lambda\rceil$). The bound of Lemma 2 follows by taking $A = [0,1]^d$ in (8). $\qquad\square$

## C    Proof of Proposition 1: original Mondrian Forests are inconsistent

In this appendix, we show that Mondrian Forests with fixed lifetime $\lambda$ are inconsistent, as stated in Proposition 1. We establish that this is true both for the variant based on the full domain $[0,1]^d$, and for the original Mondrian Forests algorithm [LRT14] that restricts to the range of training data.

### C.1    Reduction to the full domain

First, we begin by showing that, asymptotically, there is little difference between Mondrian trees constructed on the full domain and those restricted to the range of the training data. This is due to the fact that, as the sample size $n$ grows large, the training data will span the whole domain, as well as every cell contained in it.

**Lemma 3.** *Assume the distribution $\mu$ of $X$ satisfies: $\mu(A) \geqslant \alpha \operatorname{vol}(A)$ for every measurable $A \subset [0,1]^d$, for some $\alpha \in (0,1]$. Fix $\lambda > 0$. For every $n \geqslant 1$, there exists a couple $(M_\lambda, M_\lambda^{\mathrm{range}(n)})$ such that $M_\lambda \sim \mathrm{MP}(\lambda, [0,1]^d)$, $M_\lambda^{\mathrm{range}(n)}$ is a Mondrian partition with parameter $\lambda$ restricted to the range defined by the data points $X_1, \dots, X_n$, and $\mathbb{P}(M_\lambda = M_\lambda^{\mathrm{range}(n)}) \to 1$ as $n \to \infty$.*

*Proof of Lemma 3.* Let $M_\lambda \sim \mathrm{MP}(\lambda, [0,1]^d)$ be sampled by the procedure `SampleMondrian` (Algorithm 1). We will define explicitly each $M_\lambda^{\mathrm{range}(n)}$ so that they have the desired distribution, and agree with $M_\lambda$ on an event of high probability.

First, consider the event $\Omega_n$ that all splits of $M_\lambda$ occur inside the range defined by the feature points among $X_1, \dots, X_n$ that belong to the cell to be split. We will show that $\mathbb{P}(\Omega_n) \to 1$ as $n \to \infty$. Since the tree $M_\lambda$ is grown independently of $(X_1, \dots, X_n)$, we may reason conditionally on $M_\lambda$, and $(X_1, \dots, X_n)$ remains distributed as $\mu^{\otimes n}$. Note that $\Omega_n$ is equivalent to the following: no leaf cell of $M_\lambda$ contain no points among $X_1, \dots, X_n$. We can now write, denoting $\Omega_n^c$ the complementary of $\Omega_n$,

$$\begin{aligned}
\mathbb{P}(\Omega_n^c \mid M_\lambda) &= \mathbb{P}(\exists \phi \in \mathcal{L}(M_\lambda) : A_\phi \cap \{X_1, \dots, X_n\} = \varnothing) \\
&\leqslant \sum_{\phi \in \mathcal{L}(M_\lambda)} \mathbb{P}(A_\phi \cap \{X_1, \dots, X_n\} = \varnothing) \\
&= \sum_{\phi \in \mathcal{L}(M_\lambda)} (1 - \mu(A_\phi))^n \\
&\leqslant \sum_{\phi \in \mathcal{L}(M_\lambda)} (1 - \alpha \operatorname{vol}(A_\phi))^n \tag{9} \\
&\underset{n \to \infty}{\to} 0 \quad \text{a.s.} \tag{10}
\end{aligned}$$

where equation (9) used the hypothesis $\mu \geqslant \alpha \operatorname{vol}$, and the convergence (10) is almost sure with respect to $M_\lambda$, since a.s. $\operatorname{vol}(A_\phi) > 0$ for every $\phi \in \mathcal{L}(M_\lambda)$. By the dominated convergence theorem (since each random variable $\mathbb{P}(\Omega_n^c \mid M_\lambda)$, $n \geqslant 1$, is dominated by 1), we have $\mathbb{P}(\Omega_n) = \mathbb{E}[\mathbb{P}(\Omega_n^c \mid M_\lambda)] \to 0$ as $n \to \infty$.

For every $n \geqslant 1$, we define $M_\lambda^{\mathrm{range}(n)}$ as follows: on $\Omega_n^c$, we let $M_\lambda^{\mathrm{range}(n)}$ be a random Mondrian partition of lifetime $\lambda$, on the range defined by the data points $X_1, \dots, X_n$. On $\Omega_n$, we take

$M_\lambda^{\text{range}(n)}$ to be a pruning of $M_\lambda$. Specifically, for $\eta \in \mathcal{N}(M_\lambda)$, denote $E_\eta = E_{A_\eta} \sim \text{Exp}(|A_\eta|)$ the exponential random variables drawn during the construction of $M_\lambda$ (see Algorithm 1). Now, set $E_\eta^{\text{range}(n)} := \frac{|A_\eta|}{|A_\eta^{\text{range}(n)}|} E_\eta \sim \text{Exp}(|A_\eta^{\text{range}(n)}|)$, and $\tau_\eta^{\text{range}(n)} := \sum_{\eta'} E_{\eta'}^{\text{range}(n)}$, where the sum spans over the (strict) ancestors $\eta' \in \mathcal{N}(M_\lambda)$ of $\eta$. Finally, we define $M_\lambda^{\text{range}(n)}$ on $\Omega_n$ to be equal to the pruning of $M_\lambda$ obtained by keeping only the nodes $\mathcal{N}$ such that $\tau_\eta^{\text{range}(n)} \leqslant \lambda$. By construction, $M_\eta^{\text{range}(n)}$ has the distribution of a Mondrian process of parameter $\lambda$ restricted to the range of the data $X_1, \ldots, X_n$.

It remains to show that $\mathbb{P}(M_\lambda^{\text{range}(n)} = M_\lambda) \to 1$. Since we already proved that $\mathbb{P}(\Omega_n) \to 1$, it suffices to show that $\mathbb{P}(M_\lambda^{\text{range}(n)} = M_\lambda \mid \Omega_n) \to 1$.

Second, consider the random variable $\Delta_n = \sup_{\phi \in \mathcal{L}(M_\lambda)} \frac{|A_\phi|}{|A_\phi^{\text{range}(n)}|} - 1 \geqslant 0$. By the same argument as above, but replacing the boxes $A_\phi$ ($\phi \in \mathcal{L}(M_\lambda)$) by interior cubes of size $\varepsilon$ around the edges of the cells $A_\eta$ ($\eta \in \mathcal{N}(M_\lambda)$), we see that $\Delta_n \to 0$ in probability as $n \to \infty$. Since a.s. $\tau_\phi < \lambda$ and $\tau_\phi^{\text{range}(n)} \leqslant (1 + \Delta_n)\tau_\phi$ for every $\phi \in \mathcal{L}(M_\lambda)$, we have $\mathbb{P}(M_\lambda^{\text{range}(n)} = M_\lambda \mid \Omega_n) \to 1$, which concludes the proof. $\qquad\square$

## C.2 A simple example for fixed lifetime and range

In order to establish Proposition 1, it remains to provide a simple counter-example that proves the inconsistency of the Mondrian Forest algorithm for a fixed range and lifetime.

*Proof.* Fix $\lambda > 0$, and let $\epsilon \in (0, \frac{1}{4})$ to be specified later. Let $X$ be uniformly distributed on $[0,1]$; we set $Y = 1$ if $|X - \frac{1}{2}| \leqslant \epsilon$, and $0$ otherwise. Clearly, we have $L^* = 0$.

Denote $\widehat{g}_{\lambda,n}^{(K)}$ the classifier described in Algorithm 4 with $\lambda_n = \lambda$, trained on the dataset $((X_1, Y_1), \ldots, (X_n, Y_n))$, and denote $\widehat{\eta}_{\lambda,n}^{(K)}$ the corresponding estimate of the conditional probability $\eta$. Also, let $M_\lambda \sim \text{MP}(\lambda, [0,1]^d)$ and denote $A_\lambda(x) \subset [0,1]$ the cell of $x \in [0,1]$, as well as

$$N_{\lambda,n}(x) := \sum_{i=1}^n \mathbf{1}_{\{X_i \in A_\lambda(x)\}}, \qquad \widehat{\eta}_{\lambda,n}(x) := \frac{1}{N_{\lambda,n}(x)} \sum_{i=1}^n Y_i \cdot \mathbf{1}_{\{X_i \in A_\lambda(x)\}}$$

(with $\widehat{\eta}_{\lambda,n}(x) = 0$ if $N_{\lambda,n}(x) = 0$) and $\widehat{g}_{\lambda,n}(x) := \mathbf{1}_{\{\widehat{\eta}_n(x) \geqslant \frac{1}{2}\}}$. For each $x \in [\frac{1}{2} - \epsilon, \frac{1}{2} + \epsilon]$, we have

$$\mathbb{P}(\widehat{g}_{\lambda,n}^{(K)}(x) = 1) = \mathbb{P}(\widehat{\eta}_{\lambda,n}^{(K)}(x) \geqslant 1/2) \leqslant 2\,\mathbb{E}[\widehat{\eta}_{\lambda,n}^{(K)}(x)] = 2\,\mathbb{E}[\widehat{\eta}_{\lambda,n}(x)]$$

by Markov's inequality and the fact the $K$ trees in the forest have the same distribution as $M_\lambda$. Now, conditionally on $A_\lambda(x)$ and on $N_{\lambda,n}(x) = N \geqslant 1$, the points among $X_1, \ldots, X_n$ that fall in $A_\lambda(x)$ are $N$ i.i.d. points drawn uniformly in the interval $A_\lambda(x)$, and $\widehat{\eta}_{\lambda,n}(x)$ is just the fraction of those points that satisfy $|X_i - \frac{1}{2}| \leqslant \epsilon$. In particular,

$$\mathbb{E}[\widehat{\eta}_{\lambda,n}(x) \mid A_\lambda(x), N_{\lambda,n}(x) = N] = \frac{|A_\lambda(x) \cap [1/2 - \epsilon, 1/2 + \epsilon]|}{|A_\lambda(x)|} \leqslant \frac{2\epsilon}{|A_\lambda(x)|}$$

so that

$$\mathbb{P}(\widehat{g}_{\lambda,n}^{(K)}(x) = 1) \leqslant 2\epsilon\,\mathbb{E}[|A_\lambda(x)|^{-1}]. \tag{11}$$

Now, recall that $M_\lambda$ is a partition of $[0,1]$ into subintervals whose endpoints form a Poisson point process of intensity $\lambda$ (Fact 2). In particular, a direct derivation shows that $\mathbb{E}[|A_\lambda(x)|^{-1}] \leqslant F(\lambda) := \lambda + 4e^{-\lambda/4} < +\infty$. Choosing $\epsilon := \frac{1}{4} \wedge \frac{1}{4F(\lambda)}$ and using Equation (11), we get $\mathbb{P}(\widehat{g}_{\lambda,n}^{(K)}(x) = 1) \leqslant \frac{1}{2}$. Finally, integrating over $X$, we get for each $n \geqslant 1$:

$$L(g_n^{(K)}) \geqslant \int_{1/2 - \epsilon}^{1/2 + \epsilon} \mathbb{P}(\widehat{g}_{\lambda,n}^{(K)}(x) = 0)dx \geqslant \epsilon > 0, \tag{12}$$

so that $L(g_n^{(K)})$ is bounded away from 0, as announced. $\qquad\square$

# D   Proof of Theorem 1: consistency for Mondrian forests

## D.1   Some general consistency results

Let us recall two general consistency results that will be used in the proof. First, the consistency of Mondrian forests can be deduced from that of the individual trees, using Proposition 2.

**Proposition 2** (Proposition 1 in [BDL08]). *If a sequence $(\widehat{g}_n)_{n \geqslant 1}$ of randomized classifiers is consistent, then for each $K \geqslant 1$, the averaged classifier $\widehat{g}_n^{(K)}$ is consistent.*

Then, to establish the consistency of individual trees, we use the following consistency theorem for partitioning classifiers.

**Proposition 3** ([DGL96], Theorem 6.1). *Consider a sequence of randomized tree classifiers $(\widehat{g}_n(\cdot, Z))$, grown independently of the labels $Y_1, \dots, Y_n$. For $x \in [0,1]^d$, denote $A_n(x) = A_n(x, Z)$ the cell containing $x$, $\operatorname{diam} A_n(X)$ its diameter, and $N_n(x) = N_n(x, Z)$ the number of input vectors among $X_1, \dots, X_n$ that fall in $A_n(x)$. Assume that, if $X$ is drawn from the distribution $\mu$:*

1. *$\operatorname{diam} A_n(X) \to 0$ in probability, as $n \to \infty$,*

2. *$N_n(X) \to \infty$ in probability, as $n \to \infty$,*

*Then, the tree classifier $\widehat{g}_n$ is consistent.*

## D.2   Universal consistency

We will need Lemma 4 which states that the number of training observations in the cell of a point tends to infinity with $n$, if the number of splits is controlled.

**Lemma 4.** *Assume that the total number of splits $K_{\lambda_n}$ performed by the Mondrian tree partition $M_{\lambda_n}$ satisfies $\mathbb{E}(K_{\lambda_n})/n \to 0$. Then, $N_n(X) \to \infty$ in probability.*

*Proof.* The proof extends a result in [BDL08] to a random number of splits. We fix $n \geqslant 1$, and reason conditionally on $M_{\lambda_n}$, which is by construction independent of $\mathscr{D}_n$ and $X$. Note that the number of leaves is $|\mathcal{L}(M_{\lambda_n})| = K_{\lambda_n} + 1$, and let $(A_\phi)_{\phi \in \mathcal{L}(M_{\lambda_n})}$ be the corresponding cells. For $\phi \in \mathcal{L}(M_{\lambda_n})$ we define $N_\phi$ to be the number of points (with repetition) among $X_1, \dots, X_n, X$ that fall in the cell $A_\phi$. Since $X_1, \dots, X_n, X$ are i.i.d., so that the joint distribution of $(X_1, \dots, X_n, X)$ is invariant under permutation of the $n+1$ points, conditionally on the set $S = \{X_1, \dots, X_n, X\}$ (and on $M_{\lambda_n}$) the probability that $X$ falls in the cell $A_\phi$ is $\frac{N_\phi}{n+1}$. Therefore, for each $t > 0$,

$$\mathbb{P}(N_n(X) \leqslant t) = \mathbb{E}\{\mathbb{P}(N_n(X) \leqslant t \mid S, M_{\lambda_n})\}$$

$$= \mathbb{E}\left\{ \sum_{\phi \in \mathcal{L}(M_{\lambda_n}) \,:\, N_\phi \leqslant t} \frac{N_\phi}{n+1} \right\}$$

$$\leqslant \mathbb{E}\left\{ \frac{t|\mathcal{L}(M_{\lambda_n})|}{n+1} \right\}$$

$$= \frac{t(\mathbb{E}(K_{\lambda_n}) + 1)}{n+1},$$

which tends to 0 as $n \to \infty$ by assumption. $\qquad\square$

*Proof of Theorem 1.* To prove the consistency of Mondrian forest with a lifetime sequence, we show that the two assumptions of Proposition 3 are satisfied, which proves Theorem 1 since our algorithm performs splits independently of the labels $Y_1, \dots, Y_n$. First, Lemma 1 ensures that, if $\lambda_n \to \infty$, $D_{\lambda_n}(x) = \operatorname{diam} A_{\lambda_n}(x) \to 0$ in probability for every $x \in [0,1]^d$. In particular, for every $\delta > 0$, $\mathbb{P}(\operatorname{diam} A_{\lambda_n}(X) \geqslant \delta) = \int_{[0,1]^d} \mathbb{P}(\operatorname{diam} A_{\lambda_n}(x) \geqslant \delta)\mu(dx) \to 0$ as $n \to \infty$ by the dominated convergence theorem. This establishes the first condition.

For the second condition, Lemma 2 implies that $\mathbb{E}(K_{\lambda_n})/n \leqslant e^d(\lambda_n + 1)^d/n \to 0$ by hypothesis. By Lemma 4, this establishes the second condition of Lemma 3, which concludes the proof. $\qquad\square$

# E   Proof of Theorem 2: Minimax rates for Mondrian forests in regression

In this section, we demonstrate how the properties about Mondrian trees established in Lemmas 1 and 2 imply minimax rates over the class of Lipschitz regression function, in arbitrary dimension $d$. We consider the following regression problem

$$Y = f(X) + \varepsilon,$$

where $X$ is a $[0,1]^d$-valued random variable, $\varepsilon$ is a real-valued random variable such that $\mathbb{E}(\varepsilon \mid X) = 0$ and $\mathrm{Var}(\varepsilon \mid X) \leqslant \sigma^2 < \infty$ a.s., and $f : [0,1]^d \to \mathbf{R}$ is $L$-Lipschitz. We assume to be given $n$ i.i.d. observations $(X_1, Y_1), \ldots, (X_n, Y_n)$, distributed as $(X, Y)$. We draw $K$ i.i.d. Mondrian tree partitions $M_{\lambda_n}^{(1)}, \ldots, M_{\lambda_n}^{(K)}$, distributed as $\mathrm{MP}(\lambda_n, [0,1]^d)$. For all $k = 1, \ldots, K$, we let $\widehat{f}_n^{(k)}(x)$ be the $k$th Mondrian tree estimate at $x$, that is the average[1] of the labels $Y_i$ such that $X_i$ belongs to the cell containing $x$ in the partition $M_{\lambda_n}^{(k)}$. Finally, the Mondrian forest estimate at $x$ is given by

$$\widehat{f}_n = \frac{1}{K} \sum_{k=1}^K \widehat{f}_n^{(k)} : [0,1]^d \to \mathbf{R} \,.$$

**Proposition 4.** *The quadratic risk $R(\widehat{f}_n) = \mathbb{E}(\widehat{f}_n(X) - f(X))^2$ of $\widehat{f}_n$ is upper bounded as follows:*

$$R(\widehat{f}_n) \leqslant \frac{4dL^2}{\lambda_n^2} + \frac{1 + e^d(1 + \lambda_n)^d}{n} \left(2\sigma^2 + 9\|f\|_\infty\right) \tag{13}$$

*In particular, the choice $\lambda_n = n^{1/(d+2)}$ yields a risk rate $R(\widehat{f}_n) = O(n^{-2/(d+2)})$.*

*Proof.* First, by the convexity of the function $y \mapsto (y - f(x))^2$ for any $x \in [0,1]^d$, we have $R(\widehat{f}_n) \leqslant \frac{1}{K} \sum_{k=1}^K R(\widehat{f}_n^{(k)}) = R(\widehat{f}_n^{(1)})$ since the random trees classifiers have the same distribution. Hence, it suffices to prove the risk bound (13) for a single tree; in the following, we assume that $K = 1$, and consider the random estimator $\widehat{f}_n$ associated to a tree partition $M_{\lambda_n} \sim \mathrm{MP}(\lambda_n, [0,1]^d)$.

Since the splits of the tree partition $M_{\lambda_n}$ are performed independently of the training data $(X_1, Y_1), \ldots, (X_n, Y_n)$ we can write the following bias-variance decomposition of the risk for *purely random forests*, first noticed by [Gen12]:

$$R(\widehat{f}_n) = \mathbb{E}(f(X) - \tilde{f}_{\lambda_n}(X))^2 + \mathbb{E}(\tilde{f}_{\lambda_n}(X) - \widehat{f}_{\lambda_n}(X))^2 \,, \tag{14}$$

where we denoted $\tilde{f}_{\lambda_n}(x) := \mathbb{E}(f(X)|X \in A_{\lambda_n}(x))$ (which only depends on the random partition $M_{\lambda_n}$) for every $x$ in the support of $\mu$. The first term of the sum, the *bias*, measures how close $f$ is to its best approximation $\tilde{f}_n$ that is constant on the leaves of $M_{\lambda_n}$ (on average over $M_{\lambda_n}$). The second term (the *variance*) measures how well the expected value $\tilde{f}_n(x) = \mathbb{E}(Y \mid X \in A_{\lambda_n}(x))$ (*i.e.* the optimal label on the leaf $A_{\lambda_n}(x)$) is estimated by the empirical average $\widehat{f}_n(x)$, averaged over the sample $\mathscr{D}_n$ and the partition $M_{\lambda_n}$.

The bias term is bounded as follows: for each $x \in [0,1]^d$ in the support of $\mu$, we have

$$
\begin{aligned}
|f(x) - \tilde{f}_n(x)| &= \left| \frac{1}{\mu(A_{\lambda_n}(x))} \int_{A_{\lambda_n}(x)} (f(x) - f(z))\mu(dz) \right| \\
&\leqslant \sup_{z \in A_{\lambda_n}(x)} |f(x) - f(z)| \\
&\leqslant L \sup_{z \in A_{\lambda_n}(x)} \|x - z\|_2 \\
&= LD_{\lambda_n}(x),
\end{aligned} \tag{15}
$$

where $D_{\lambda_n}(x)$ is the $\ell^2$-diameter of $A_{\lambda_n}(x)$; note that inequality (15) used the assumption that $f$ is $L$-Lipschitz. By Lemma 1, this implies

$$\mathbb{E}(f(x) - \tilde{f}_n(x))^2 \leqslant L^2 \mathbb{E}[D_{\lambda_n}(x)^2] \leqslant \frac{4dL^2}{\lambda_n^2} \,. \tag{16}$$

Integrating the bound (16) with respect to $\mu$ yields the following bound on the integrated bias:

$$\mathbb{E}(f(X) - \tilde{f}_n(X))^2 \leqslant \frac{4dL^2}{\lambda_n^2} . \tag{17}$$

In order to bound the variance term, we use the following fact ([AG14], Proposition 2): if $U$ is a random tree partition of the unit cube in $k + 1$ cells (with $k \in \mathbf{N}$ deterministic) formed independently of the training data $\mathscr{D}_n$, we have

$$\mathbb{E}(\tilde{f}_U(X) - \widehat{f}_U(X))^2 \leqslant \frac{k+1}{n} \left(2\sigma^2 + 9\|f\|_\infty\right) . \tag{18}$$

For every $k \in \mathbf{N}$, applying the upper bound (18) to the random partition $M_{\lambda_n} \sim \mathrm{MP}(\lambda_n, [0,1]^d)$ conditionally on the event $\{K_{\lambda_n} = k\}$ (where $K_{\lambda_n}$ denotes the number of splits performed by $M_{\lambda_n}$), and summing over $k$, we get

$$\begin{aligned}
\mathbb{E}(\tilde{f}_{\lambda_n}(X) - \widehat{f}_{\lambda_n}(X))^2 &= \sum_{k=0}^{+\infty} \mathbb{P}(K_{\lambda_n} = k)\, \mathbb{E}[(\tilde{f}_{\lambda_n}(X) - \widehat{f}_{\lambda_n}(X))^2 \mid K_{\lambda_n} = k] \\
&\leqslant \sum_{k=0}^{+\infty} \mathbb{P}(K_{\lambda_n} = k)\, \frac{k+1}{n} \left(2\sigma^2 + 9\|f\|_\infty\right) \\
&= \frac{1 + \mathbb{E}(K_{\lambda_n})}{n} \left(2\sigma^2 + 9\|f\|_\infty\right) .
\end{aligned}$$

Then, applying Lemma 2 gives an upper bound of the variance term:

$$\mathbb{E}(\tilde{f}_{\lambda_n}(X) - \widehat{f}_{\lambda_n}(X))^2 \leqslant \frac{1 + e^d(1 + \lambda_n)^d}{n} \left(2\sigma^2 + 9\|f\|_\infty\right) . \tag{19}$$

Combining the bounds (17) and (19) with the decomposition (14) yields the desired bound (13). $\qquad\square$

## Footnotes

[1] With the convention that if no training point $X_i$, $1 \leqslant i \leqslant n$, falls in $A_{\lambda_n}(x)$, then $\tilde{f}_n(x) := 0$.