[Reviews · NeurIPS 2017]

Reviewer 1



Summary: This paper proposes a modification of Mondorian Forest which is a variant of Random Forest, a majority vote of decision trees. The authors show that the modified algorithm has the consistency property while the original algorithm does not have one. In particular, when the conditional probability function is Lipschitz, the proposed algorithm achieves the minimax error rate, where the lower bound is previously known. Comments: The technical contribution is to refine the original version of the Mondorian Forest and prove its consistency. The theoretical results are nice and solid. The main idea comes from the original algorithm, thus the originality of the paper is a bit incremental. I’m wondering that in Theorem 2, there is no factor about K, the number of Mondrian trees, as authors mentioned in Section 5. In my intuition, it is natural that many K results more accurate and converges faster. Ultimately, Theorem 2 says the Mondrian forest works well even if K=1. Is there any theoretical evidence of advantage by using many Mondrian trees? Computational time of Algorithm 4 is not clear. It seems to me that increasing lifetime parameters leads more time consumption in SplitCell. In theorem 2, it is natural to measure the difference between two distributions by KL divergence. Do you have any idea of using another discrepancy measure instead of the quadratic loss? Additionally, the proof, the variance of noise \sigma assumed to be a constant. Is it a natural setting? Minor Comments: Line 111 g_n(x,Z_k) -> \hat{g}_n(x,Z_k) Line 117 \nu_{d_\eta} -> \nu_{\eta} Line 3 of pseudocode of Algorithm 2 \tau_A -> \tau Line 162 M_{\lambda^{\prime}} \sim MP(\lambda,C) -> M_{\lambda^{\prime}} \sim MP(\lambda^{\prime},C)

Reviewer 2



In this paper, the authors study the consistency of an algorithm of Mondrian Forest. First, they propose to use an increasing lifetime parameters in Mondrian Forest, and develop an updating rule that works in online setting. Then, they establish simple conditions for consistency, and also prove that their algorithm achieves the minimax rate for the estimation of a Lipschitz regression function. Strong Points: 1. This paper is well-motivated. 2. The authors present conditions for consistency of their modified Mondrian Forest, which is novel. 3. The convergence rate is minimax optimal, which is quite strong. Limitations: 1. A potential limitation is the explicit dependence on the dimensionality $d$. 2. There are no empirical studies of their modified algorithm. Questions: 1. The results in this paper do not have any dependence on the number of trees $K$. Does it mean we can set $K=1$? If this is true, why do we need forest? A single tree is enough. 2. What value of $\lambda_n$ should we use in practice?

Reviewer 3



This paper consider the consistency of a (online) variant of Mondrian Forest. The authors amend the original Mondrian Forest algorithm where a fixed lifetime parameter hinders statistical consistency of the original procedure. The authors modify the modified Mondrian Forest algorithm grows trees with increasing lifetime parameters, and uses an alternative updating rule, allowing to work also in an online fashion. Then, the authors prove the statistical consistency of the modified Mondrian Forest algorithm, and achieve the minimax rate (optimal rate) for the estimation of a Lipschitz regression function. This work is generally well-written, while I have the following concerns: 1) The proposed online algorithm is a modification of original Mondrian Forest with different parameter, and the statistical consistency depends on the choices on the parameter. How about the performance on real datasets? I think the authors should present some real experiments to show the effectiveness of the proposed online learning algorithm, especially for the generalization of the proposed algorithm. 2) It is easy to prove the main statistical consistency and the minimax rate without new technical insights from theoretical view.